# Primer Choice and Xylem-Microbiome-Extraction Method Are Important Determinants in Assessing Xylem Bacterial Community in Olive Trees

**DOI:** 10.3390/plants11101320

**Published:** 2022-05-16

**Authors:** Manuel Anguita-Maeso, Carmen Haro, Juan A. Navas-Cortés, Blanca B. Landa

**Affiliations:** Department of Crop Protection, Institute for Sustainable Agriculture, Spanish National Research Council (CSIC), 14004 Córdoba, Spain; charo@ias.csic.es (C.H.); j.navas@csic.es (J.A.N.-C.)

**Keywords:** microbiome, endophytes, xylem, PCR primers, NGS

## Abstract

Understanding the unique and unexplored microbial environment of xylem sap is starting to be of relevant importance for plant health, as it could include microbes that may protect plants against xylem-limited pathogens, such as *Verticillium dahliae* and *Xylella fastidiosa*. In this study, we evaluated the effects that the method for extracting the xylem bacterial communities, the plant age and the PCR primers may have on characterizing the xylem-bacterial-community composition by using an NGS approach. Xylem sap was extracted from xylem vessels by using a Scholander pressure chamber, or by macerating wood shavings that were obtained from xylem tissues by using branches from 10-year-old olive trees, or the entire canopy of 1-year-old olive plantlets. Additionally, we compared four different PCR-primer pairs that target 16S rRNA for their efficacy to avoid the coamplification of mitochondria and chloroplast 16S rRNA, as this represents an important drawback in metabarcoding studies. The highest amplifications in the mitochondria and chloroplast reads were obtained when using xylem woody chips with the PCR1-799F/1062R (76.05%) and PCR3-967F/1391R (99.96%) primer pairs. To the contrary, the PCR2-799F/1115R and PCR4-799F/1193R primer pairs showed the lowest mitochondria 16S rRNA amplification (<27.48%), no chloroplast sequences and the highest numbers of bacterial OTUs identified (i.e., 254 and 266, respectively). Interestingly, only 73 out of 172 and 46 out of 181 genera were shared between the xylem sap and woody chips after amplification with PCR2 or PCR4 primers, respectively, which indicates a strong bias of the bacterial-community description, depending on the primers used. Globally, the most abundant bacterial genera (>60% of reads) included *Anoxybacillus*, *Cutibacterium*, *Pseudomonas*, *Spirosoma*, *Methylobacterium-Methylorubrum* and *Sphingomonas*; however, their relative importance varied, depending on the matrix that was used for the DNA extraction and the primer pairs that were used, with the lowest effect due to plant age. These results will help to optimize the analysis of xylem-inhabiting bacteria, depending on whether whole xylematic tissue or xylem sap is used for the DNA extraction. More importantly, it will help to better understand the driving and modifying factors that shape the olive-xylem-bacterial-community composition.

## 1. Introduction

The olive tree (*Olea europaea* L.) is one of the most emblematic crop species in the Mediterranean Basin, which is due to its ecological, economic and cultural importance [1,2,3]. Currently, the health status of olive trees is being threatened by a remarkable increase in diseases that are caused by xylem-inhabiting pathogens, such as the plant pathogenic bacterium *Xylella fastidiosa*, and the soilborne fungus *Verticillium dahliae*. Both pathogens are global threats for olive production worldwide, as they are capable of adversely affecting olive growth and production, which causes huge economic losses and has severe environmental impacts [4,5,6,7,8]. 

Plants have evolved diverse mechanisms to defend themselves against plant pathogens. Thus, bacterial and fungal endophytes that have beneficial relationships with their host plants may colonize xylem vessels and other internal plant tissues, which evokes a series of key responses that have beneficial effects on plant growth and/or may confer protection against pathogens [9]. The presence of these microbial endophytes, and their effects on the host plant, should be addressed in more detail because of the potential beneficial effects that these microorganisms might have if used as biocontrol agents [10].

The study of the plant microbiome has received considerable attention over the last decades because it is a key determinant factor in plant health and productivity [11]. Having more information on the plant microbiome may rebound in alternatives to potentially solve some of the environmental issues of pesticides through the enhancement of the plant and soil health and, as a consequence, the plant productivity [12]. Furthermore, some members of the plant-associated microbiome may be exploited as biocontrol agents that provide protection against plant pathogens, and especially those of difficult management, such as soilborne and vascular pathogens [13,14]. While there is a huge number of studies that focus on the description of the soil- and root-associated microbiome, only a few publications have focused on identifying the diversity in the xylem microbiome and its role in plant health and productivity [15,16,17,18,19,20,21]. The lack of comprehensive studies on this plant niche may be due, in part, to the technical difficulties that are involved in the isolation of the xylem-inhabiting microorganism and/or because of the microbiome complexity itself, which can also be affected by several biotic (such as the plant age and the host cultivar or variety) and abiotic (agronomic factors, climate, seasonality, etc.) conditions [14,22,23,24,25]. The accurate and nonbiased identification of xylem-inhabiting microorganisms could be a determinant for plant protection against vascular pathogens, such as *X. fastidiosa* and *V. dahliae.* Thus, some specific bacterial taxa that are natural inhabitants of the xylem vessels could be exploited as biocontrol agents that act through the direct inhibition of these pathogens through antibiosis, competition for space, or by inducing the activation of the plant immune system [14,26].

Next-generation sequencing approaches that are based on the amplification of the 16S rRNA gene in bacteria are cost-effective and fast tools that can provide valuable phylogenetic information for the comparison of bacterial-community composition, and especially in experiments where a large number of samples are analyzed within a short timeframe. However, microbial characterization that is based on the PCR amplification of 16S rRNA from plant-associated matrices can be particularly problematic because of the chloroplast- and mitochondria-sequence contamination [27,28]. Additionally, the results of bacterial profiling can vary considerably according to the hypervariable region that is amplified within the 16S rRNA, which depends on the choice of primers [21,29]. Multiple primer pairs are available for 16S rRNA analysis; consequently, each pair needs to be carefully selected and tested on specific samples to avoid taxonomy biases in the microbial-community analysis, and to allow for a comparison with different datasets [21,30]. Overall, although specific primer pairs have been recommended for plant-microbiome studies [31,32,33], when using samples with substantial plant-associated tissue contamination, the choice of primers has to be carefully considered and tested in order to avoid the unspecific amplification of the mitochondrial and chloroplast sequences [21,34,35]. Several specific mismatch primers, which amplify bacterial 16S rDNA sequences while simultaneously avoiding the amplification of organellar DNA sequences, have been developed with that purpose, and they reveal different performances, depending on the study [20,21,36,37].

In a previous study [21], we determined the strong effects of the DNA-extraction protocol and of several primer pairs targeting different 16 rRNA regions when assessing the bacterial microbiota profile of olive xylem sap that was extracted by using a Scholander chamber. The extraction of xylem sap from olive with this device is not always feasible, and especially when a large number of samples need to be assessed in a short period of time, or when the sampling sites are located at long distances from laboratory facilities, due to the high negative stem-water potential of olive trees [38,39]. Consequently, other methods for extracting xylem microbiome, such as the maceration of xylematic tissue, can be used [20]. Indeed, this is the most widely used approach for studies that target xylem microbiome. However, the experimental performances of different primers, and their efficacies in reducing the coamplification of olive DNA when using macerated tissues compared to sap fluids, have not been evaluated sufficiently.

This work was focused on establishing a reliable method for the characterization of microbial communities that are present in the xylem of olive trees. We demonstrated that both the choice of the PCR-primer pair and the xylem-associated microbiome-extraction procedure have strong effects on the metabarcoding analysis when assessing the structure and diversity of the olive-xylem-bacterial-community composition. We also describe significant differences in the xylem-associated bacterial communities according to the olive age. The knowledge that is gathered in this work concerning the effects of these factors may be fundamental for large-scale microbiome studies that are aimed at identifying the microbial populations of this rarely addressed plant niche, and it minimizes the biases that are caused by the presence of nonspecific amplifications when using NGS approaches.

## 2. Materials and Methods

### 2.1. Plant Sampling

Nursery-propagated plantlets (seedlings) (1-year-old) and adult (10-years-old) olive trees were used in the study. Adult olive trees were sampled from an experimental field at the Institute for Sustainable Agriculture from the Spanish National Research Council (IAS-CSIC) in Córdoba (Southern Spain). Olive plantlets grew in the same location for six months before sampling. A total of 18 terminal 35 cm-long branches from six independent trees (three branches per tree) and six independent olive plantlets (entire plantlet) were used for xylem extraction. Plants were processed in October, due to the stability of the olive stem-water potential in this season that facilitates xylem-sap extraction through the use of the Scholander chamber device [40]. Parallel 6 cm-long portions from each branch (one woody piece per branch in adult trees, and three pieces per olive plantlets) were selected for xylematic-tissue extraction. All pruned branches were placed in sterile plastic bags, sprayed with distilled water and kept in a cold room at 4 °C to avoid desiccation until sample processing within the same day.

Xylem-sap extraction from xylem vessels or from macerated woody chips was carried out as described in Anguita-Maeso et al. (2020) [20]. Briefly, a 1505D-EXP Scholander pressure chamber (PMS Instruments, Albany, OR, USA) was used to perform xylem-sap extraction from xylem vessels with an external port that allows the inclusion of branches up to 60 cm long. After inserting the branch into the super chamber, 2 cm of the main stem was debarked and disinfested with a sterile paper moistened in ethanol to avoid the microbial contamination of the xylem sap from bark and phloem. Ethanol was left to evaporate before sap extraction. The pressure was increased gradually until xylem-sap drops were observed, but to a maximum of 35 bars of pressure to avoid external contamination derived from cell rupture [20]. On the other hand, short stem portions were debarked, and xylem chips were obtained by scraping the most external layers of the debarked woody pieces with a sterile scalpel. Xylem chips were mixed, and a 0.5 g sample was placed in an extraction Bioreba bag (BIOREBA, Reinach, Switzerland) containing 5 mL of sterile phosphate-buffered saline (PBS); the bags were closed with a thermal sealer, and the contents were macerated with a hand homogenizer (BIOREBA). All extracts were stored at −80 °C until DNA extraction [20].

### 2.2. DNA Extraction and Library Sequencing

DNA isolation from xylem-sap samples extracted with the Scholander chamber and from macerated xylem chips was performed by using the DNeasy PowerSoil Kit (QIAGEN, Hilden, Germany), with a sample pretreatment step prior, to follow the DNA manufacturer’s protocol. Briefly, aliquots of xylem samples (0.5 mL) obtained from macerated xylem chips were placed directly in PowerBead tubes, whereas xylem-sap samples extracted with the Scholander chamber were filtered through a 0.22 µm-pore MF-Millipore™ filter (Merck Millipore, Madrid, Spain). Then, extracts were homogenized 7 min at 50 pulses s^−1^ with the Tissuelyser LT (QIAGEN), and were incubated in the lysis buffer for 1 h at 60 °C to increase cell lysis [20]. DNA was eluted in a final volume of 50 μL of ultrapure, filter-sterilized distilled water, the purity of the DNA (absorbance 260/280 nm ratio) was determined by using a NanoDrop^®^156 ND-1000 UV-Vis spectrophotometer (Thermo Fisher Scientific, Inc., Waltham, MA, USA) and the yield concentration was quantified by using the Quant-iT^TM^ PicoGreen^TM^ dsDNA Assay Kit (Thermo Fisher Scientific). This DNA was used as a template for PCR amplification with different PCR-primer combinations, which were described before, as specific mismatch primers that target different hypervariable regions of bacterial 16S rRNA, and with low affinity for organellar plant DNA (Appendix A). The primer combinations used where: PCR1-799F/1062R for the V5–V6 hypervariable region in 16S rRNA gen [41,42]; PCR2-799F/1115R for V5–V6 [43]; PCR3-967F/1391R for V6–V8 [44,45]; and PCR4-799F/1193R for V5–V7 [46]. Primer pairs PCR3-967F/1391R and PCR4-799F/1193R were used in a previous study by Haro et al. (2021) [21]. In that study, PCR4-799F/1193R was selected as the primer pair that provides better results for describing the bacterial community in olive xylem sap. However, these primer pairs have not been tested yet by using macerated xylem tissues.

Each PCR included 4 µL of MyTaq Reaction Buffer (5X), 0.2 µL of forward and reverse primers (10 µM), 1.3 µL of DMSO, 3 µL of genomic DNA template, 0.2 µL of MyTaq™ DNA Polymerase (5 U/µL) (Bioline Laboratories, London, UK) and 11.10 µL of ddH_2_O. The PCR-amplification program started with 5 min at 95 °C, then 35 cycles (60 s, 95 °C; 45 s, X-°C; 60 s, 72 °C) and 8 min at 72 °C. The OligoCalc online resource was used to identify the annealing temperature for each PCR primer tested [47]. Aliquots (5 µL) of PCR products were loaded onto a 1% agarose gel to confirm 16S rDNA amplification.

After successful amplification, a second PCR was performed to include identifier barcodes in each sample. For that, the PCR product (5 µL of 1/100 diluted PCR) obtained previously and 2 µL of Fluidigm barcodes (Access Array™ Barcode Library for Illumina^®^ Sequencers—384, Single Direction; San Francisco, CA, USA) were added to a PCR premix, which included 5 µL of MyTaq Reaction Buffer (5X), 1.3 µL of DMSO, 0.2 µL of MyTaq™ DNA Polymerase (5 U/µL) (Bioline Laboratories) and 11.50 µL of ddH_2_O. The PCR amplification conditions started with 5 min at 95 °C, then 7 cycles (60 s, 95 °C; 30 s, 65 °C; 60 s, 72 °C) and 8 min at 72 °C. Aliquots (5 µL) of PCR products were loaded onto a 1% agarose gel to confirm the barcode addition.

PCR products were purified by using Agencourt AMPure XP (Beckman Coulter Inc., Brea, CA, USA), following the manufacturer’s instructions. Additional purification steps were needed in PCR4, which consisted of cutting out the desired DNA band of 438 bp from the agarose gel by using Cut&Spin Gel Extraction Columns (GRiSP Research Solutions, Porto, Portugal), since an additional band of 800 bp that corresponds to plant-mitochondria-amplification products was frequently coamplified [36]. Purified PCR products were quantified by using the Quant-iT™ PicoGreen™ dsDNA Assay Kit (Thermo Fisher Scientific) and a TECAN SAFIRE microplate reader (Tecan Group, Männedorf, Switzerland). Equimolecular amounts from each individual sample were added to a single tube; the pooled library was quantified by using a 2100 Bioanalyzer (Agilent, Santa Clara, CA, USA), and was purified again if primer dimers were still evident. Finally, the library was sequenced on the Illumina MiSeq platform (V2; PE 2× 250 bp) at the Genomics Unit at the Madrid Science Park, Madrid, Spain. The ZymoBIOMICS microbial standard (Zymo Research Corp., Irvine, CA, USA) and water (no template DNA) were used as internal positive and negative controls, respectively, for library construction and sequencing. Individual fastq files have been deposited in the Sequence Read Archive (SRA) database at the NCBI under BioProject accession number PRJNA826322.

### 2.3. Bioinformatics and Statistical Approximation

After obtaining Illumina MiSeq raw sequences, FastQC and TrimGalore tools were applied to fastq files for quality control and adapter trimming. No truncation or trimming length of the forward and reverse reads was needed due to the high Phred-quality (Q > 30) score visualized in the MultiQC tool. Quality sequences were analyzed and classified by using DADA2 and VSEARCH pipelines integrated into QIIME2 (version 2019.10; https://view.qiime2.org/; accessed on 10 April 2020) [48,49]. DADA2 was applied to denoise fastq paired-end sequences and filter chimeras [50], while a VSEARCH consensus taxonomy classifier was used to obtain operational taxonomic units (OTUs) against the Silva SSU v.138 database at 99% similarity [51]. Singletons were discarded for downstream analysis.

The alpha diversity as the OTUs’ richness, Shannon and Simpson diversity and Faith_pd phylogenetic diversity, as well as rarefaction curves, were calculated to find differences in the bacterial-community composition according to the olive-plant material and plant age in each PCR-primer combination. Additionally, principal coordinate analysis (PCoA) using weighted UniFrac distances at the OTUs’ level was used to evaluate the phylogenetic distances among bacterial communities [52]. Alpha and beta diversity, as well as alpha rarefaction curves, were conducted by rarefying all samples to the minimum number of reads found. Rarefaction curves were performed by using the “iNEXT” package in R [53].

A Kruskal–Wallis test (*p* < 0.05) with FDR correction [54] was used to find differences in alpha-diversity indices among the studied factors, and the PERMANOVA test (*p* < 0.05) was used to test for significant differences, according to the olive-plant age or xylem-sap extract in each PCR-primer combination. Differential taxonomical abundances found in xylem extracted with the Scholander chamber or woody-chip maceration, plantlets or adult olive trees and PCR-primer combinations, were identified by using proportional Venn diagrams generated with the “eulerr” R package [55], which visualized the shared (core microbiome) or unique taxa in each factor studied. Heat trees summarizing the taxonomical main results, from phyla to OTUs, were created using the “Metacoder” package in R software [56]. Statistical redundancy analysis based on variation partitioning in distance-based RDA (db-RDA) using the “vegan” package in R software was applied to determine the likelihood of olive-plant age and xylem-sap-extraction method in each PCR tested as sets of predictors in explaining patterns in microbial-community structures [57].

## 3. Results

### 3.1. Illumina Output and Taxa Diversity of Plastids and Bacterial Community

The MiSeq sequencing analysis reported a total of 274,794 good-quality reads, which were distributed among 63,273 reads for PCR1-799F/1062R (23.03%), 94956 for PCR2-799F/1115R (34.55%), 50,706 for PCR3-967F/1391R (18.45%) and 65,859 for PCR4-799F/1193R (23.97%), and which were retained for the microbiome analysis after the removal of chimeras and unassigned reads. The taxonomic affiliation indicated the highest percentage of mitochondria in PCR1-799F/1062R (56.47%), whereas the lowest percentage was found in PCR2-799F/1115R (7.85%). No mitochondria reads were detected in PCR3-967F/1391R, nor in PCR4-799F/1193R. Moreover, the highest level of chloroplast reads was detected in PCR3-967F/1391R (91.18%), while no chloroplast reads were detected for the rest of the primer combinations (Figure 1). In PCR1-799F/1062R and PCR2-799F/1115R, the highest percentage of mitochondria reads was found in xylem sap that was extracted by the woody-chip-maceration method, as compared to the Scholander-chamber extraction (PCR1-799F/1062R: 76.05% vs. 27.48%; PCR2-799F/1115R: 14.11% vs. 1.43%), while a higher value of mitochondria reads was found in adult trees, as compared to olive plantlets (PCR1-799F/1062R: 75.79% vs. 43.75%; PCR2-799F/1115R: 16.60% vs. 1.48%). Similarly, the percentage of chloroplast reads followed the same tendency in PCR3-967F/1391R. Greater values of chloroplast reads were found when using macerated woody chips and in adult trees (99.65% and 99.96%, respectively) than with the Scholander-chamber method and in plantlet plants (54.67% and 86.43%, respectively) (Figure 1).

A total of 419 OTUs were identified for all of the treatments after the singleton removal. Similar numbers of OTUs were identified in PCR2-799F/1115R and PCR4-799F/1193R (254 and 266, respectively), followed by PCR1-799F/1062R (150) and PCR3-967F/1391R (26). In PCR1-799F/1062R, PCR2-799F/1115R and PCR3-967F/1391R, the highest numbers of OTUs (101, 181 and 16, respectively) were identified on xylem sap that was obtained with the Scholander-chamber method, as compared to the woody-chip-maceration method (93, 164 and 14, respectively), while, in PCR4-799F/1193R, the opposite trend occurred (72 vs. 248). Furthermore, PCR1-799F/1062R and PCR2-799F/1115R showed the highest numbers of OTUs in the xylem sap that was extracted with the Scholander chamber in adult trees (72 and 131, respectively), whereas, in PCR3-967F/1391R, this occurred for the xylem-sap samples that were extracted with the Scholander chamber in olive plantlets (14), and, in PCR4-799F/1193R, it occurred in the xylem sap that was extracted by using the macerated woody chips of adult trees (186) (Figure 1).

A total of 82 OTUs were retained after rarefying all of the data to 536 sequences (minimum reads obtained in one of the samples), along with the singleton-read and organellar-read removals. The maximum values of a Good’s coverage of 1.0 were obtained for all samples, which indicates enough sequencing coverage (data not shown). Globally, the OTUs’ richness, which is represented by the rarefaction curves, showed significant differences according to the PCR primers that were used (H = 43.919; *p* < 0.001), with a significantly higher number of OTUs in the xylem sap that was extracted with the woody-chip-maceration method, as compared to the Scholander-chamber method (H = 4.672; *p* = 0.031). When analyzing the data by PCR-primer pair, PCR4-799F/1193R presented significant differences in the OTUs’ richness according to the xylem-extraction method (H = 16.86; *p* < 0.001), whereas PCR1-799F/1062R, PCR2-799F/1115R and PCR3-967F/1391R did not (H < 2.51; *p* > 0.112). No significant differences were found between plantlets and adult olive trees (H = 0.117; *p* = 0.733) (Appendix A). The OTUs’ alpha-diversity indices (richness, Shannon, Simpson and Faith_pd) showed global significant differences among the PCR-primer combinations (H < 43.919, *p* < 0.001) and the xylem-sap-extraction methods (H > 4.541, *p* < 0.033). Significant differences were not found between the olive-plant ages (H < 2.493, *p* > 0.114) (Figure 2 and Appendix A).

### 3.2. Xylem-Bacterial-Community Distribution

A total of 17 phyla, 35 classes, 79 orders, 140 families and 269 genera were identified, considering all of the treatments and after the removal of the mitochondria and chloroplast reads. PCR2-799F/1115R and PCR4-799F/1193R showed the highest number of phyla (14), followed by PCR1-799F/1062R (12) and PCR3-967F/1391R (5). Similarly, the highest numbers of genera were identified in PCR2-799F/1115R and PCR4-799F/1193R (172 and 181, respectively), followed by PCR1-799F/1062R (113) and PCR3-967F/1391R (20). This trend was observed for other taxonomic levels, where PCR2-799F/1115R and PCR4-799F/1193R detected similar numbers of classes, orders and families, followed by PCR1-799F/1062R and PCR3-967F/1391R (Figure 3).

Five phyla were detected in all of the PCRs (Actinobacteriota, Bacteroidota, Firmicutes, Fusobacteriota and Proteobacteria), while the Planctomycetota phylum was only detected in PCR1-799F/1062R, and the Spirochaetota and Nitrospirota phyla in PCR4-799F/1193R (Figure 3). At the genus level, a total of 16 genera formed the core microbiome (shared genera) among all of the PCR-primer pairs analyzed (*Anoxybacillus*, *Arthrobacter*, *Brevibacillus*, *Brevundimonas*, *Corynebacterium*, *Cutibacterium*, *Massilia*, *Methylobacterium-Methylorubrum*, *Neisseria*, *Nocardioides*, *Pseudomonas*, *Ralstonia*, *Rathayibacter*, *Sphingomonas*, *Sphingopyxis* and *Spirosoma*). The highest numbers of unique genera were detected in PCR2-799F/1115R and PCR4-799F/1193R (53 and 68, respectively), in contrast to PCR1-799F/1062R, with 23 unique genera detected, and PCR3-967F/1391R, where no unique genera were detected. In addition, 107 (39.78%) genera were identified in both PCR2-799F/1115R and PCR4-799F/1193R, and 73 (27.14%) genera were shared by PCR1-799F/1062R, PCR2-799F/1115R and PCR4-799F/1193R. With regard to the xylem-microbiome-extraction method, a total of 111 (41.26%) genera were shared for both extraction methods, and 91 (33.83%) genera were identified in both adult and plantlet olive plants (Figure 4A); however, some differences were found according to the primers used. Thus, the xylem sap that was extracted with the Scholander chamber from adult trees showed the highest numbers of unique genera in PCR1-799F/1062R and PCR2-799F/1115R (30 and 45, respectively), whereas only one unique genus was detected in PCR3-967F/1391R, and six unique genera in PCR4-799F/1193R. Remarkably, PCR4-799F/1193R detected the highest numbers of unique genera in adult plants and plantlets from woody-chip extraction (74 and 25, respectively), whereas this variation in the plant age was not detected for the remaining combinations (Figure 4B).

Totals of 173 and 207 genera were detected in the xylem sap that was extracted with the Scholander chamber and when using macerated woody chips, respectively (Appendix A). Only eight bacterial genera (3.86% of the total identified) were shared among all PCRs when using the woody-chip-extraction procedure (*Arthrobacter*, *Corynebacterium*, *Cutibacterium*, *Methylobacterium-Methylorubrum*, *Nocardioides*, *Rathayibacter*, *Sphingomonas* and *Spirosoma*), and 10 shared genera (5.78%) were detected when using the Scholander-chamber methodology (*Anoxybacillus*, *Brevibacillus*, *Corynebacterium*, *Cutibacterium*, *Massilia*, *Methylobacterium-Methylorubrum*, *Pseudomonas*, *Ralstonia*, *Rathayibacter* and *Sphingopyxis*). Four genera (*Corynebacterium*, *Cutibacterium, Methylobacterium-Methylorubrum* and *Rathayibacter*) were detected in both methods. Interestingly, PCR2-799F/1115R showed the greatest number of unique genera (68) for the Scholander-chamber-extraction method, followed by PCR1-799F/1062R (22) and PCR4-799F/1193R (15), while PCR4-799F/1193R showed the highest number of unique genera (78) when using woody chips, followed by PCR2-799F/1115R (25) and PCR1-799F/1062R (8) (Appendix A). When comparing PCRs for each olive-plant age, a total of 132 genera were detected in plantlets, whereas 228 were found in adult trees. Only two bacterial genera (0.88% of the total identified) were shared among all of the PCRs within adult trees (*Corynebacterium* and *Cutibacterium*), which is in contrast to the 15 bacterial genera (11.36%) that were found in plantlets (*Anoxybacillus*, *Arthrobacter*, *Brevibacillus*, *Corynebacterium*, *Cutibacterium*, *Massilia*, *Methylobacterium-Methylorubrum*, *Neisseria*, *Nocardioides*, *Pseudomonas*, *Ralstonia*, *Rathayibacter*, *Sphingomonas*, *Sphingopyxis* and *Spirosoma*). PCR4-799F/1193R revealed the greatest number of unique genera (30) in plantlets, followed by PCR2-799F/1115R (23) and PCR1-799F/1062R (6). Likewise, PCR4-799F/1193R showed the highest number of unique genera (64) in adult olive trees, followed by PCR2-799F/1115R (52) and PCR1-799F/1062R (20). No unique genera were detected in PCR3-967F/1391R, irrespective of the factor of the study that was analyzed (Appendix A).

### 3.3. Bacterial-Abundance Analysis

At the phylum level, Firmicutes presented the highest relative abundance, considering all of the PCR-primer pairs used (39.41%), followed by Proteobacteria (29.88%), Actinobacteriota (23.27%), Bacteroidota (6.70%) and Abditibacteriota (0.41%). However, these relative abundances varied within each PCR tested, although the five most abundant phyla were placed in the same order for all of the PCRs. Firmicutes reached the maximum relative proportions in PCR3-967F/1391R (84.34%) and PCR1-799F/1062R (51.11%), whereas similar relative percentages were detected in PCR2-799F/1115R and PCR4-799F/1193R (35.97% and 36.02%, respectively). Globally, Proteobacteria (29.17%) and Actinobacteriota (22.65%) showed similar relative percentages in all of the PCRs, except in PCR3-967F/1391R, in which they showed the minimum relative percentages (13.94% and 1.48%, respectively). Firmicutes was the most abundant phyla in the xylem sap that was extracted with the Scholander chamber in olive plantlets (81.47%). It was remarkable the lack of Actinobacteriota in the xylem sap of plantlets that were extracted with the Scholander chamber (1.34%), in comparison to its relative abundance in all of the treatments studied (43.82%) (Appendix A).

Globally, at the genus level, *Anoxybacillus* (31.01%), *Cutibacterium* (11.67%), *Methylobacterium-Methylorubrum* (5.15%), *Pseudomonas* (5.10%), *Spirosoma* (3.64%), *Sphingomonas* (3.41%), *Rathayibacter* (3.40%) and *Massilia* (2.54%) were the most abundant genera. Thus, *Anoxybacillus* was the genus with the highest relative abundance, considering all the PCRs, and it reached the maximum frequency of detection in PCR3-967F/1391R (84.07%). Then, the genera showing the highest relative proportions varied, depending on the PCR-pair combination. Thus, *Cutibacterium* (7.15%) and *Methylobacterium-Methylorubrum* (4.17%) were the most representative genera in PCR1-799F/1062R, *Cutibacterium* (13.86%) and *Spirosoma* (6.26%) in PCR2-799F/1115R, *Pseudomonas* (6.82%) and *Massilia* (3.92%) in PCR3-967F/1391R and *Cutibacterium* (11.47%) and *Pseudomonas* (6.82%) in PCR4-799F/1193R. Noticeable differences in the relative percentages of abundance were found for *Cutibacterium* from the xylem sap that was extracted with woody-chip maceration in adult trees (32.32%), compared to those found in the rest of the treatments (15.32%) (Figure 5).

### 3.4. Effects of PCR-Primer Pairs, Xylem-Extraction Method and Plant Age on Bacterial-Community Structure

A principal-coordinate analysis of the weighted UniFrac distances of the OTU-level differentiated xylem-bacterial communities was conducted, and mainly according to the PCR-primer pairs that were used for the 16S rRNA gene amplification. This approach showed a clear trend to group the bacterial communities first by PCR-primer combination, followed by the xylem-extraction method and, finally, by the olive-plant age (Figure 6 and Appendix A). Thus, the PERMANOVA indicated a significant clustering that was due to the PCR primers that were used (*pseudo*-*F* = 22.169; *p* = 0.001), the xylem-extraction method (*pseudo*-*F* = 4.771; *p* = 0.001) and the olive-plant age (*pseudo*-*F* = 4.841; *p* = 0.001).

## 4. Discussion

The olive tree is a crop of vast ecological, economic and cultural significance for the Mediterranean Basin, which must be sustained for the successive generations [1]. In the last years, it has been proposed that plant-associated microorganisms could help to maintain olive health and productivity throughout the decades, which would require a better understanding of their community structure and diversity, and the factors that shape them [10,23]. Recently, several studies that use amplicon-based NGS approaches have contributed to a more complete characterization of the composition of plant-associated microbial communities (plant microbiome), and have allowed us to address questions that concern their potential functions in the modulation of the host properties, and particularly in nutrient acquisition and the defense against plant pathogens [58]. However, some methodological challenges when assessing plant-associated microbial communities by NGS approaches may arise, which are mainly due to mitochondria and chloroplast coamplification from the host, which may bias the results that are obtained [59]. To establish a standardized methodological approach for assessing the xylem-associated microbiome, and to overcome this undesired coamplification, in this study, we evaluated four PCR-primer pairs that target different regions of 16S rRNA, and we determined the effect that the method that is used for extracting the microbiome from the xylem sap (i.e., the use of the Scholander-chamber device or woody-chip maceration) may exert on this approach. Finally, we assessed how these factors could interact when determining the role of the plant age (plantlets vs. adult trees) in shaping the xylem microbiome structure.

Firstly, we observed that a different proportion of mitochondria reads were coamplified with the primers PCR1-799F/1062R and PCR2-799F/1115R, while a high amplification of chloroplast resulted when the primers PCR3-967F/1391R were used. The fact that the PCR4-799F/1193R primers did not amplify the organellar reads might be explained by the fact that the amplified products were further purified by cutting the desired amplicon band after the agarose-gel electrophoresis; whereas, for the rest of the PCR-primer pairs that were tested, this additional step was not performed. Furthermore, a high number of mitochondria reads were observed when the woody-chip-maceration procedure was used, and especially from adult trees, in contrast to xylem-sap extraction with the Scholander chamber, and mainly for plantlets. Similarly, the frequency of the chloroplast detection followed the same tendency when using PCR3-967F/1391R.

We also found a significant influence of the primer-pair selection when estimating the bacterial alpha diversity, as has been reported in previous studies [36,60], where PCR2-799F/1115R and PCR4-799F/1193R reached higher values of diversity indices (richness, Shannon, Simpson and Faith_pd), as compared to PCR1-799F/1062R and PCR3-967F/1391R. Consequently, primer pairs 799F/1115R (PCR2) and 799F/1193R (PCR4) performed better for the assessment of the bacterial diversity after the removal of the plastid reads (92.15% and 100% of bacterial reads, respectively). Nevertheless, we recommend the use of the primer pairs 799F/1115R (PCR2) for plant-microbiome studies, as it recovers a higher bacterial diversity with a reasonably low percentage of mitochondria reads, and with the advantage that the additional purification step, based on the excision of the expected amplification band after agarose-gel electrophoresis, is not necessary.

Our taxonomic assignment results indicate the presence of a high relative abundance (>1%) of endophytic bacteria from the phyla Actinobacteriota, Bacteroidota, Firmicutes, Fusobacteriota and Proteobacteria for all of the PCR-primer pairs tested, while the phylum Planctomycetota was only observed in PCR1-799F/1062R, and the phyla Spirochaetota and Nitrospirota were only detected in PCR4-799F/1193R. This fact evidences a taxonomy bias that depends on the PCR-primer-pair selection. The use of primer 1062R has been associated with the amplification of Planctomycetota, in accordance with other studies [61,62], while the phyla Spirochaetota and Nitrospirota are specifically detected when amplifying the V7 hypervariable region with the primer 1193R [63,64,65]. These detected phyla (Firmicutes, Proteobacteria, Actinobacteriota and Bacteroidota) were ordered in the same way, according to their decreasing relative abundances for each of the PCR-primer-pair combinations that were used, although the relative values changed according to the primer pair that was used.

Firmicutes was the most abundant phylum in the olive xylem sap that was extracted with the Scholander chamber, and the low abundance of Actinobacteriota in the olive xylem sap from plantlets that was extracted with the Scholander chamber was noticeable. These results are in agreement with those obtained by Fausto et al. (2018) [17], which found similar phyla proportions by using this same device for xylem-sap extraction. The main bacterial genera that were detected in our study, as indicated by their high relative abundances in all of the PCR-primer combinations, were *Anoxybacillus*, *Cutibacterium*, *Methylobacterium-Methylorubrum*, *Pseudomonas*, *Spirosoma*, *Sphingomonas*, *Rathayibacter* and *Massilia*. Interestingly, these eight genera were present for both extraction methods, and independently of the olive-plant age (except for *Spirosoma*, which was detected only in plantlets). These results indicate that these genera are highly abundant and stable, and that they could be members of the olive xylem core microbiome. Slight differences in the relative abundance of the genus *Anoxybacillus* were found when comparing the different PCR primers (reaching the maximum levels in PCR3-967F/1391R). The genus *Cutibacterium* also exhibited a high relative abundance, and its presence as a main component of xylem sap has been reported by several authors [66,67]. Finally, noticeable differences in the relative abundance of *Cutibacterium* was found when the xylem sap was extracted with the woody-chip-maceration procedure in adult trees (32.32%), compared to the rest of the treatments (15.32%). This fact may be due to the location of this bacterium into the xylem vessels, or in surrounding tissues in adult trees, and its release was favored by the tissue-maceration process; although, more studies are needed to unravel this hypothesis.

Independently of the primer pairs that were used, 16 genera (i.e, *Anoxybacillus, Arthrobacter, Brevibacillus, Brevundimonas, Corynebacterium, Cutibacterium, Massilia, Methylobacterium-Methylorubrum, Neisseria, Nocardioides, Pseudomonas, Ralstonia, Rathayibacter, Sphingomonas, Sphingopyxis* and *Spirosoma*) formed the core microbiome. The detection of these genera in xylem sap may indicate that they are common inhabitants of the xylem vessels, as they have also been detected in previous studies in olive [14,20,21], and in other woody plants, such as poplar, walnut, grapes, *Pinus* and *Platanus* [68,69,70,71,72,73]. Additionally, four of these 16 genera (*Corynebacterium, Cutibacterium, Methylobacterium-Methylorubrum* and *Rathayibacter*) were detected by both xylem-sap-extraction methods and all four PCR-primer pairs, which is in accordance with others studies that have also detected these genera associated with xylem tissues [74,75,76]. However, *Corynebacterium* and *Methylobacterium-Methylorubrum* have also been detected in other plant niches, such as in roots, leaves and soils [77,78,79], which may serve as source or sink niches.

Secondly, we found that the xylem-extraction method that was used (Scholander chamber vs. woody chips) had an important effect on the characterization of the xylem microbiome diversity, since it affected the identification of specific taxa in each extraction method. In fact, in this study, we detected 34 OTUs more when using the macerated-tissue-extraction method than when using the Scholander-chamber device. This higher number in macerated tissue might be explained by the extraction of some microorganisms that may occupy intercellular tissues that surround the xylem vessels and that could not be true inhabitants of the vascular tissues.

The use of macerated xylematic tissue is the most widely used approach in studies that target xylem microbiome, which is due to the fact that this technique is more accessible, less tedious and laborious with an effective time-consuming protocol, and more affordable (no need to use a nitrogen tank) than the use of the Scholander device [20,24]. Furthermore, our results indicate that the effects of the primer pairs and the extraction method were stronger when using olive plantlets, whereas, for adult olive trees, the bacterial-community composition was less affected by these factors, which indicates that both extraction methods may be used equally when adult trees are sampled.

Thirdly, we found differences in the core bacterial composition according to the olive-plant age when all of the PCR-primer combinations were considered (only two genera were shared among all PCRs in adult trees, in contrast to 15 in plantlets). These results indicate that the plant age has an undoubted effect on the microbial-community structure and assembly, which agrees with other reports [80,81,82]. Several studies have indicated differences in the microbial-community composition according to plant age, but this has been mainly addressed in herbaceous species (i.e., mustard, potato, *Arabidopsis* and soybean) [80,82,83,84], and much less in woody crops [85,86], and only two studies have been conducted on olive [23,87]. Although there are several publications that have focused on the characterization of olive-xylem-inhabiting microorganisms that use culture-independent approaches [14,16,17,18,19,20,88], only one of our previous studies [23] analyzed the effect of the plant age on the assembling of these communities. In this study, our results show significant differences in the microbiome-community composition that are associated with the xylem sap, depending on the olive-tree age, which indicates more the diverse bacterial composition of adult olive trees as compared to plantlets, as was indicated by the higher number of total and unique OTUs that were identified in adult trees, independently of the primer pairs that were used. Although this work shows some evidence of the existence of differences in the microbial-community structure according to the olive age, a more targeted experimental design is needed to explore how plant-associated microbial communities are maintained or are specifically selected as the olive-tree ages. The fact that olive trees can select a stable microbiota during plant growth or development may be a crucial finding in studies that are aimed at improving the plant-health status and the disease resistance to vascular pathogens, and especially if some selected microorganisms are to be inoculated through endotherapy treatments that start from the nursery stage during vegetative propagation [23,89].

Although several studies report the importance of the adequate selection of PCR-primer pairs to avoid the coamplification of mitochondria and chloroplast reads in plant-microbiome analysis when sequencing 16S rRNA [36,60], our study reveals, for the first time, the important influence of the appropriate choice of PCR-primer pairs and xylem-extraction procedure on the characterization of the structure and diversity of the xylem-bacterial-community composition that is based on the amplification of the 16S rRNA. We found that the primer pair 799F-1115R (PCR2) provides a higher depth and taxa coverage compared to the other primer pairs that were tested, with a low percentage of mitochondria coamplification, but with the benefit of being a simpler approach, as it does not need an additional purification step (i.e., the excision of the PCR-amplification products from agarose gels). To conclude, we recommend the use of the primer pair 799F-1115R for olive microbiome research in order to minimize the biases that are caused by the presence of nonspecific amplifications when using NGS approaches. Our results will be relevant for future studies that address the identification and characterization of the olive xylem microbiome and its determining factors. The standardization of this NGS approach will be very useful for deciphering the core xylem-inhabiting bacteria from olive with the most stable populations (i.e., less affected by environmental factors or the host genotype of plant aging). This standardization represents a first critical step in the identification of the potential members to be isolated and then tested as biocontrol agents against vascular plant pathogens, which contributes to the support of the health and growth of the olive tree.

## Figures and Tables

**Figure 1 plants-11-01320-f001:**
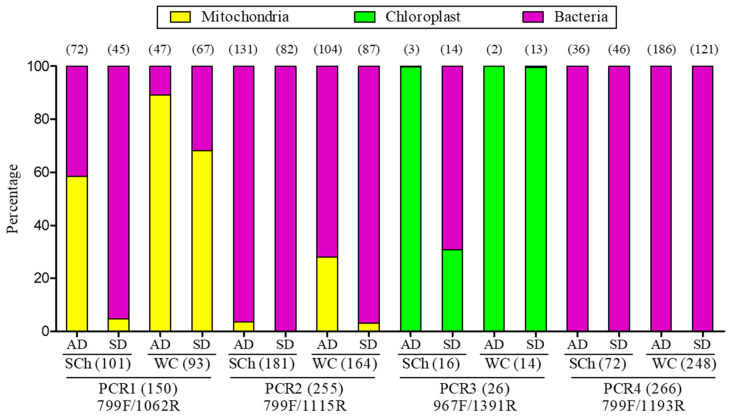
Relative abundances of mitochondria, chloroplast and bacteria in xylem sap of plantlets (SD) or adult (AD) olive plants extracted with Scholander chamber (SCh) or woody chips (WC) as determined by using four PCR-primer combinations. Number of observed OTUs are indicated between brackets.

**Figure 2 plants-11-01320-f002:**
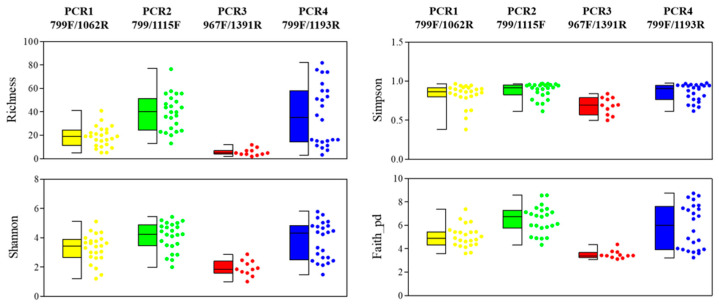
Richness, Simpson, Shannon and Faith_PD diversity indices at OTUs’ taxonomic level determined by using four different PCR-primer pairs. Dots represent the values for all samples tested in each PCR-primer pairs.

**Figure 3 plants-11-01320-f003:**
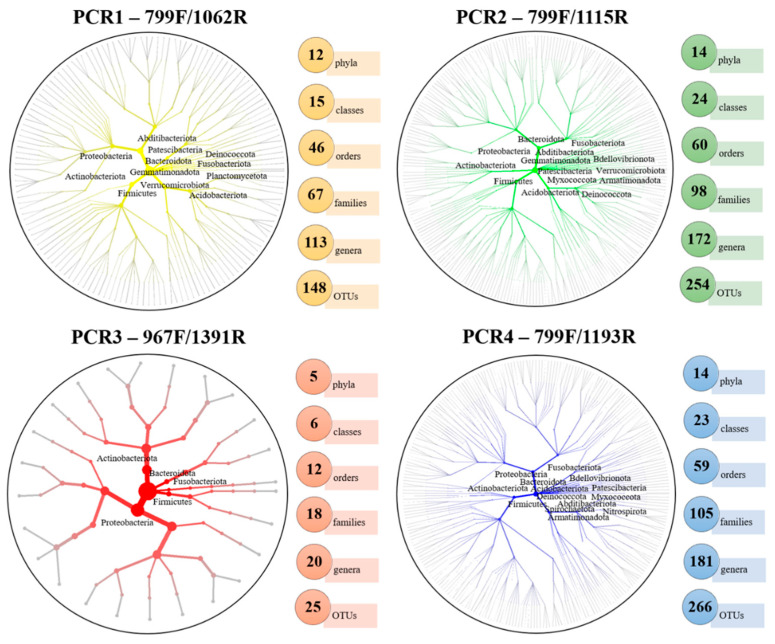
Heat trees and main taxonomic ranks identified from phyla to OTU level for each PCR-primer combination tested. Identified phyla are shown within each heat tree.

**Figure 4 plants-11-01320-f004:**
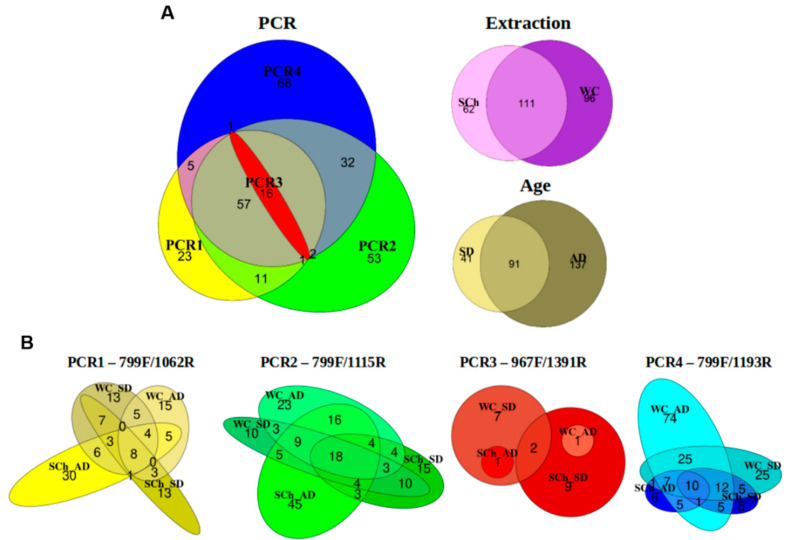
Proportional elliptical Venn diagrams showing the unique and shared bacterial genera obtained in xylem sap of plantlets (SD) or adult (AD) olive plants extracted by using the Scholander chamber (SCh) or macerated woody chips (WC), and for each PCR-primer pair used. (**A**) Bacterial genera identified for each factor of the study. (**B**) Bacterial genera identified according to the plant age and xylem-extraction method shown for each PCR tested.

**Figure 5 plants-11-01320-f005:**
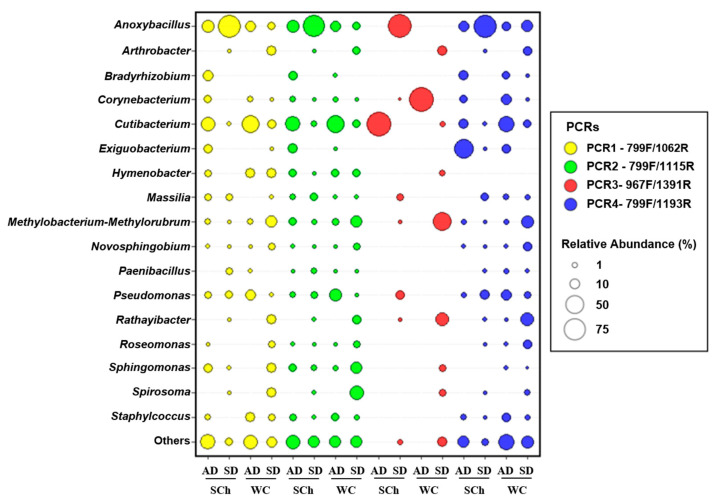
Taxonomic bubble plot of bacterial relative abundance at genus level present in olive xylem sap and identified for each PCR-primer combination of plantlets (SD) or adult (AD) olive plants extracted with the Scholander chamber (SCh) or from woody-chip maceration (WC). Only genera with relative abundances greater than 80% of reads are shown.

**Figure 6 plants-11-01320-f006:**
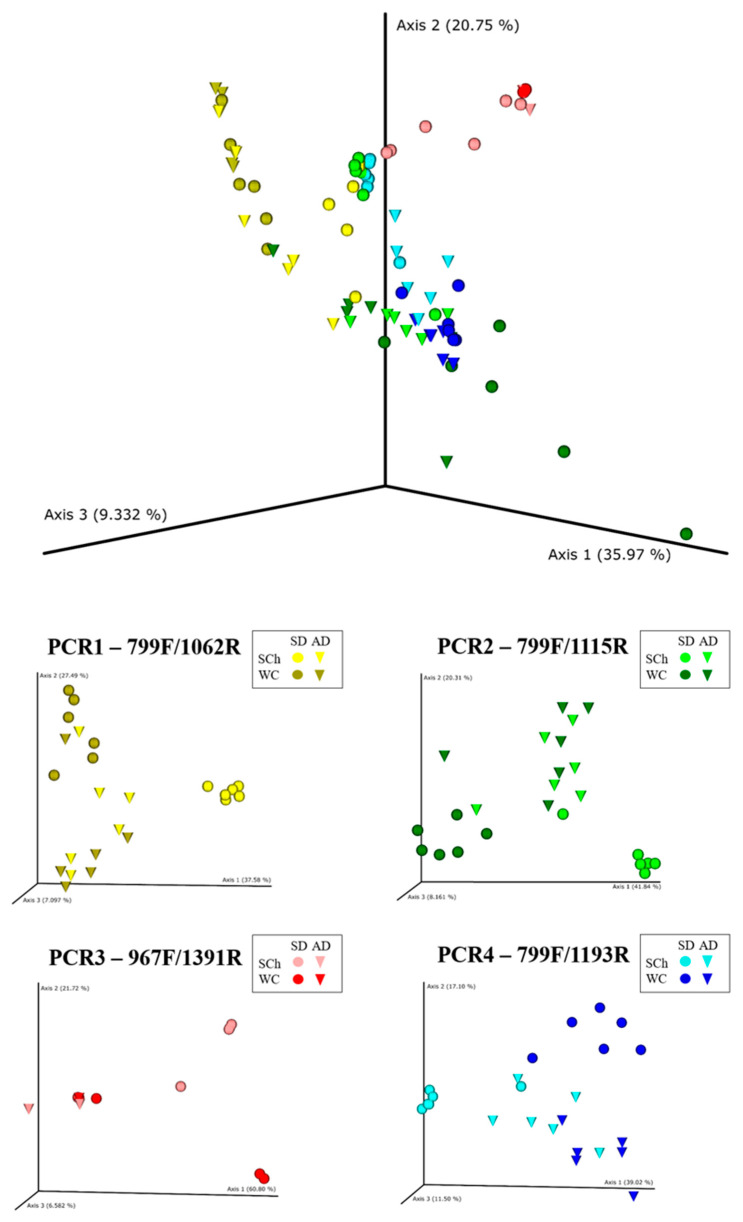
Principal-coordinate plots of weighted UniFrac distances of bacterial communities at OTUs’ taxonomic level in xylem sap of plantlets (SD) or adult (AD) olive plants extracted with the Scholander chamber (SCh) or woody chips (WC) within each PCR-primer combination tested. Points are colored by PCR-primer pairs, where xylem-sap-extraction method is represented by color intensity and olive-plant age by shape.

## Data Availability

The raw sequence data have been deposited in the Sequence Read Archive (SRA) database at the NCBI under BioProject accession number PRJNA826322.

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
