# Peer review of "Primer Choice and Xylem-Microbiome-Extraction Method Are Important Determinants in Assessing Xylem Bacterial Community in Olive Trees"

_plants, 2022, doi:10.3390/plants11101320_

Round 1
Reviewer 1 Report
The manuscript by Anguita-Maeso et al. describes the comparison of four different primer pairs, two extraction methods and two seedling ages in the 16S metabarcode analysis of the xylem microbiome of olive. The authors demonstrate that targeting of specific hypervariable regions of the 16S gene could reduce the co-amplification of plant organellar DNA, thereby greatly increasing the proportion of bacterial taxa identified. They also demonstrated these different source materials, extraction methods and primer pairs each amplified a considerable number of differing bacterial taxa from the same material. However, while there were clear differences in taxa amplified via these various methods, the most abundant genera were found to be consistent between methods. The authors propose that these abundant and consistent genera could be considered a core olive xylem microbiome which may be of value in future design of biocontrol approaches against vascular pathogens.
Although largely descriptive, the manuscript describes novel findings that will be of interest to all those working with 16S metabarcode analysis of plant microbiomes, particularly those working with the difficulties of analysing endophytes of woody tissue. It will also be interest to those looking to design biocontrol approaches against vascular pathogens in woody plants. It is well-written and the reasoning is well-constructed and clear. The methods are well-described, the data are convincing, the discussion is sound and the conclusions reached are justified.
The following points need to be addressed, though:
Title: “Primer choice and xylem microbiome extraction method are important determinants …” rather than “Primer choice and xylem microbiome extraction method are main determinants …”
Line 19: “On the contrary” rather than “On the opposite”
Line 58: “only a few publications” rather than “only a few number of publications”
Line 272/273: While this is true overall, this is largely down to the results of PCR4. Other primers do not show this trend. Please break this analysis down by PCR method.
Line 277: The same is true for the xylem sap method comparison. Differences are largely due to differences only observed for PCR4.
Line 310: “only one unique genus” rather than “only one genus”
Line 311: “and six unique genera in“ rather than “and six in genera in“
Line 312: I believe the sentence should read “Remarkably, PCR4-799F/1193R detected the highest number of unique genera in adult plants from woody chips extraction and plantlets from woody chips extraction (74 and 25, respectively)”. These are the two numbers given in the parentheses.
Line 322: “and 10 shared genera” rather than “and 10 genera”
Line 461: “Slight differences” rather than “Slightly differences”
Line 476: “as they have been” rather than “as them have been”
Line 493: “the most widely-used approach” rather than “the most widely approach”
Line 504: “has an undoubted effect” rather than “has an undoubtedly effect”
Line 522: “if some selected microorganisms are to be inoculated” rather than “if some selected microorganisms are thought to be inoculated”
Line 532: “with a low percentage” rather than “with an assumable low percentage”
Line 538. This final sentence is very long and convoluted and would be clearer if it could be split into two sentences.
Figure 4 and Fig S3. The black text against the dark blue background is very difficult to read
Author Response
Title: “Primer choice and xylem microbiome extraction method are important determinants …” rather than “Primer choice and xylem microbiome extraction method are main determinants …”
Corrected
Line 19: “On the contrary” rather than “On the opposite”
Corrected
Line 58: “only a few publications” rather than “only a few number of publications”
Corrected
Line 272/273: While this is true overall, this is largely down to the results of PCR4. Other primers do not show this trend. Please break this analysis down by PCR method.
These results have been incorporated into the text L274-277. Thanks for the clarification.
Line 277: The same is true for the xylem sap method comparison. Differences are largely due to differences only observed for PCR4.
This clarification has been also added to the text. Thanks.
Line 310: “only one unique genus” rather than “only one genus”
Corrected
Line 311: “and six unique genera in“ rather than “and six in genera in“
Corrected
Line 312: I believe the sentence should read “Remarkably, PCR4-799F/1193R detected the highest number of unique genera in adult plants from woody chips extraction and plantlets from woody chips extraction (74 and 25, respectively)”. These are the two numbers given in the parentheses.
Corrected. Yes, the word plantlets was missing. Thanks.
Line 322: “and 10 shared genera” rather than “and 10 genera”
Corrected
Line 461: “Slight differences” rather than “Slightly differences”
Corrected
Line 476: “as they have been” rather than “as them have been”
Corrected
Line 493: “the most widely-used approach” rather than “the most widely approach”
Corrected
Line 504: “has an undoubted effect” rather than “has an undoubtedly effect”
Corrected
Line 522: “if some selected microorganisms are to be inoculated” rather than “if some selected microorganisms are thought to be inoculated”
Corrected
Line 532: “with a low percentage” rather than “with an assumable low percentage”
Corrected
Line 538. This final sentence is very long and convoluted and would be clearer if it could be split into two sentences.
We split the final sentence into two sentences. Thanks.
Figure 4 and Fig S3. The black text against the dark blue background is very difficult to read
The colors are in accordance with the legibility of the figures. The upload of high-quality Figures will solve this problem.
Reviewer 2 Report
This is a really interesting article, since it attempts to develop a methodological protocol that minimizes, as far as possible, the problems associated with the extraction of microbial DNA in matrices of plant origin and its subsequent analysis by massive sequencing techniques. The experimental design is appropriate, and the wording of the text makes it easy to follow. In this sense, the work deserves to be published, in order to facilitate the future work of researchers working with similar objectives. Just a couple of notes:
- In the abstract it is mentioned that the effect caused by plant age is minimal compared to other factors. However, reading the text, both Results and Discussion, does not give that impression.
- L. 311-313: reference is made to the number of unique genera with the PCR4 primer pair in adult plants and with the woody chips extraction process. Supposedly, it should be one number, but two are included.
- Why does the additional step of purification of amplification products apply only to those obtained from PCR4? No reason is given for this. And if there is not, how is it possible to know to what degree it affects the level of diversity obtained compared to that which would have been obtained without such a step?
- Some cites in the text are not numerically referenced.
Author Response
In the abstract it is mentioned that the effect caused by plant age is minimal compared to other factors. However, reading the text, both Results and Discussion, does not give that impression.
In results, we mentioned that the effect caused by plant age was not significant when considered OTUs richness and alpha diversity compared with the others factors. Furthermore, although plant age resulted significant in beta diversity, F-value was very similar to xylem extraction method far behind the PCR factor. For that reason, the results support the conclusion that the highest effect was due to the PCR primers combination and xylem extraction method followed by the plant age.
311-313: reference is made to the number of unique genera with the PCR4 primer pair in adult plants and with the woody chips extraction process. Supposedly, it should be one number, but two are included.
Yes, the word plantlets was missing. Thanks. We have corrected it.
Why does the additional step of purification of amplification products apply only to those obtained from PCR4? No reason is given for this. And if there is not, how is it possible to know to what degree it affects the level of diversity obtained compared to that which would have been obtained without such a step?
In lines 182-184 we specified that Additional purification steps were needed in PCR4, that consisted of cutting out the desired DNA band from the agarose gel using Cut&Spin Gel Extraction Columns (GRiSP Research Solutions). The reason is that when performing the first PCR of the library with PCR4 primers (799F-1193R1062-799), desired amplicon bands should be 438 pb but one undesired band of 800 pb appeared. This unspecified amplification is easy to distinguish due to the size difference and will affect the useful reads obtained from Illumina platform and if maintained it will be a bias in the diversity, because you are not sequencing the target 16S amplicon proposed. So, the desired DNA band of the amplicon was selected and excised from the gel to proceed with the following steps. We have included an explanation for this in the text.
Some cites in the text are not numerically referenced.
Corrected